# Violence, Inequity, and Their Impact on Health and Access to Healthcare Services Among the Elderly Population of Bogotá

**DOI:** 10.3390/ijerph22101555

**Published:** 2025-10-13

**Authors:** Carlos Alberto Cano-Gutiérrez, Diego Andrés Chavarro-Carvajal, Julián Andrés Sucerquia-Quintero

**Affiliations:** 1Instituto de Envejecimiento de la Facultad de Medicina, Pontificia Universidad Javeriana, Bogotá 110231, Colombia; ccano@javeriana.edu.co; 2Unidad de Geriatría, Hospital Universitario San Ignacio, Bogotá 110231, Colombia; 3Red Interuniversitaria de Envejecimiento Saludable de Latinoamérica y Caribe (RIES-LAC), Pontificia Universidad Javeriana, Bogotá 110231, Colombia; 4Departamento de Psiquiatría, Facultad de Medicina, Universidad Nacional de Colombia, Bogotá 110231, Colombia; jasucerquiaq@unal.edu.co; 5Hospital Universitario Nacional de Colombia, Bogotá 110231, Colombia

**Keywords:** violence, health inequities, aged

## Abstract

Objective: This study explores the prevalence of violence and forced displacement as indicators of inequity among Bogotá’s elderly population, with a particular focus on how these factors affect their health and access to healthcare services. Methods: This is a subsidiary analysis of the SABE-Bogotá survey. The design was a probabilistic cluster sample of 2000 people aged 60 and over. The study was carried out by the Pontificia Universidad Javeriana’s Institute on Aging and cosponsored by Colciencias. The variables of interest were displacement and experiences of violence, assessed through self-reporting. A descriptive analysis of all variables was performed, calculating simple frequency distributions. Subsequently, dependency and association analyses were performed using Chi-square, T-tests, and multivariate logistic regressions, depending on each case. Results: 43.32% of the subjects were victims of some type of violence in the last year, among which offensive language was one of the most frequent. Individuals with severe depression (OR 2.10 [1.21–3.65]) and those who had been victims of displacement (OR 2.55, CI 95% [1.65–3.95]) had the highest risk of violence. The results reveal a direct correlation between these experiences and pre-existing health conditions. For instance, severe depression and a history of displacement were associated with a higher risk of experiencing violence, while the risk of displacement was higher among individuals with diabetes, severe depression, and, crucially, those who lacked access to health insurance. Conclusion: A high percentage of the elderly population in the city of Bogotá has been victims of different types of violence, including ones related to armed conflict and forced displacement, which is a particular and exclusive form of violence suffered by this group of people. These findings suggest that violence and displacement are social determinants of health that exacerbate inequities, underscoring the need for more inclusive health policies and improved access to medical care for this vulnerable population.

## 1. Introduction

Changes in demographic aging have been of great importance to humanity as tangible achievements of the development and major efforts invested in health services in general and public health in particular [1]. Life expectancy has grown considerably over the last seven decades, increasing by nearly 30 years, with a greater impact on women [2]. This process has been even more rapidly accelerated in Latin America, where people over 65 represented 6% of the population in 2010, a figure that is projected to reach 15% by 2036. This means that it will take Latin America only 26 years to double its population over the age of 65, while developed countries such as France took 115 years and the United States 69 years [3].

Colombia is at the top of the epidemiological aging curve, which is referred to as full transition, and has only been surpassed by countries that are in accelerated transition, such as Uruguay, Argentina, and Cuba [4]. In 2013, 10.5% of the population was aged 60 and over, and life expectancy is currently 79.4 for women and 73.1 for men [4].

The city of Bogotá is the most populous conurbation in the country, now a multicultural and migratory hub where people from all regions converge [5]. It had more than 7.5 million inhabitants in 2012, of whom nearly 11% were aged 60 and over, with growth rates higher than those of the country’s total population [6]. The city’s aging index was also higher than that of the country as a whole in 2010, at 39%, compared to 34.4% for the rest of the country [7].

Today, we have a better understanding of the characteristics of the elderly population, its growth, and its environment as a group. We are also more aware of their particular circumstances, ranging from specific health issues to other factors such as political, social, and cultural perspectives.

The increasing population of older adults has led to a greater understanding of their distinctive traits and requirements. This demographic shift has revealed how this group is often subject to a range of particular conditions, from specific health issues to distinct political, social, and cultural circumstances. It is crucial to understand these diverse aspects in order to develop effective policies and interventions, while also considering the need to avoid perpetuating stereotypes.

Older people are victims of different types of violence, which often go unnoticed and do not receive adequate attention in society [8]. The World Health Organization (WHO) defines violence as ‘the intentional use of physical force or power, in effect, or as a threat against oneself, another person, or against the group or community, which either results in or is likely to result in injury, death, psychological harm, maldevelopment, or deprivation’ [9]. The WHO also defines elder abuse as ‘any single or recurrent act, or lack of appropriate intervention, occurring within the context of existing or expected relationships where there is an expectation of trust, and which results in harm or distress to an older person’ [10,11]. The World Report on Violence and Health emphasizes the definition of elder abuse as a repeated act or failure to take necessary action that is associated with harm or distress to an older person [11]. Elder abuse is thus classified into the following categories: physical, psychological or emotional abuse, financial or material abuse, and sexual abuse. Older people are a vulnerable population, not only because of their age, but also because some of them have disabilities or are highly dependent on others for financial, psychological, or emotional support. The National Research Council of the United States Academies [12] adds to the above definition the absence of a caregiver to protect the elderly from harm, as well as the limited ability of that caregiver in ensuring that their basic needs are being met.

Population studies on the prevalence of abuse, violence, and harm to older people are limited and cover a range of nuances of the term ‘elder abuse’. Most often, they document recent and current events related to domestic violence in households or residential care facilities. They seek to identify risk factors [13,14,15,16] and collect information from adult social care, public health, or government agency records. These records may come from the police, emergency services, or nursing homes. Countries seeking to develop policies to deal with this issue can use this information.

It is estimated that violence and abuse against older adults are not necessarily accurately recorded, even when researchers, clinicians, or public health agencies make efforts to identify all cases. On the one hand, specific situations are not formally registered, and certain situations are not directly recognized due to a lack of knowledge of what the definition covers. It is challenging to directly estimate the social and health costs due to the underreporting of cases. Nevertheless, the few population studies available can already suggest some direct and indirect costs. For example, a recently published retrospective analysis of data obtained solely from the US National Emergency Department describes that in a single year (2012), of nearly 30,000 visits by adults aged 60 and older, nearly 4000 consultations were estimated to be associated with violence and abuse. Neglect and physical abuse were the most frequently identified causes, and the prevalence was higher in women. Although this study appears to focus on physical violence exclusively, it estimates the related costs to be over one million US dollars per year. It is important to note that this study estimates that only one in twenty-four cases is reported, which makes this public health concern even more significant [17].

Both clinicians and researchers face considerable challenges in identifying and addressing this issue. At this moment in our nation’s history, it is not only necessary but also of paramount importance to thoroughly contextualize this problem. Since poverty is one of the triggers of violence, it is important to take into account the United Nations’ proposals regarding the acquisition of resources to combat this phenomenon [18]. A subsequent review emphasizes that conflict and violence are risk factors that have been shown to significantly slow human development [19].

Armed conflicts, wars, and violence in general create public health catastrophes. Conflicts kill, maim, disable, and displace millions of people, usually to other countries, but in Colombia, this migration is also internal and has enormous implications in many different areas. In its attempt to build a peace agenda, our country must include violence and displacement not only as indicators of armed conflict but also as factors directly related to public health. The United Nations agenda rightly points out that inclusive social development goes beyond poverty eradication; Jayasinghe suggests that a specific public health goal to end wars and armed conflicts should be included in the development of the post-millennium goals [20].

According to the 2014 report of the United Nations High Commissioner for Refugees, referred to as the ‘invisible crisis in Colombia’, there were 5,840,590 displaced persons [21]. By March of 2016, the Single Victims Registry (RUV) reported that there were 7,957,219 registered victims, of whom 7,675,032 were victims of the armed conflict, and 1,602,135 were ‘direct victims of forced disappearance, homicide, or deceased and no longer active for assistance’ [22].

Within this context, political violence, as a result of the internal conflict that Colombia has been experiencing for more than 50 years, generates this ‘invisible crisis’ that is partly reflected in forced displacement due to armed violence, which affects many regions, with the city of Bogotá being a center of this crisis since its very inception [23]. The intensity and pressure of displacement due to armed violence vary greatly across the country’s departments, with Bogotá being an extreme case, as it is one of the regions that receives the highest numbers of displaced people but expels the lowest number of displaced persons. These gaps remain in the displaced population of all ages, but they are especially pronounced in the group aged 60 and over [23].

This study investigates the prevalence of violence and forced displacement as indicators of inequity within Bogotá’s elderly population. It specifically examines how these factors impact their health and access to healthcare services.

The research was conducted by the Institute on Aging at the Pontificia Universidad Javeriana and was co-financed by the Administrative Department of Science, Technology, and Innovation (Colciencias). The survey, known as SABE-Bogotá, was designed by the Pan American Health Organization in 2000 [24] and was adapted and contextualized to the particularities of our nation by our institution by including a chapter on violence, the only one to date of its kind in this type of study.

The term ‘marker’ has been used in epidemiology to refer to risk and is reserved for personal (endogenous) variables that are not controllable and define particularly vulnerable individuals. Similarly, as used in this study, this concept is applied to see and understand the characterization of certain variables associated with violence, the inclusion of displacement, and their relationship with other social and health factors, with a view to proposing possible interventions to reduce this social issue among older people in the city of Bogotá.

## 2. Materials and Methods

### 2.1. Type of Study: Analytical Cross-Sectional

Design and sampling: This is a secondary analysis of the Bogotá Health, Well-being, and Aging (SABE) Survey. Its design was a probabilistic, cluster sampling (housing segments) with stratification of entire blocks, to which a design correction factor was applied to obtain a reliability level of 95%. The sample consists of 2000 people aged 60 and over, living in private households in urban and rural areas of the city. When expanded, based on population projections for 2012 [6], there were 779,534 people aged 60 and over. The response rate was 81.9%, which supports the quality and representativeness of the study.

The instrument used in the SABE-Bogotá Survey was based on the international questionnaire used in other SABE surveys conducted in seven Latin American capitals between 1999 and 2000 [24]. The questionnaire was adapted and adjusted to the characteristics of the city without losing comparability. The research protocol was approved by the Research and Ethics Committee of the Pontificia Universidad Javeriana. Informed consent was obtained from each participant. The questionnaire was validated and adjusted through a pilot test conducted with 30 people aged 60 and over living in the city, selected by convenience sampling and considering quotas by age (‘early old age’ 60 to 69 years, and ‘late old age’ 70 years and over), sex, and socioeconomic status. Fieldwork teams were organized, consisting of one supervisor, three or four interviewers, and one anthropometrist. The teams were trained by the principal investigators, thematic researchers, a statistician, and a field coordinator. The data collected were entered and recorded in Excel for Windows. A total of 11.7% of the older adults selected in the SABE-Bogotá sample required a proxy informant to respond to the survey.

Dependent variables: The variables of primary interest were those that assessed whether individuals had been victims of violence in the last year or displacement at some point in their lives. Subsequently, the different forms of violence to which these people had been exposed were identified, and who their aggressors were. This was achieved by using the following questions in the questionnaire: ‘In the last year, have you been a victim of: offensive language, personal injury or physical assault (hitting, slapping, kicking), sexual abuse, robbery, kidnapping, or none of the above?’ ‘Who were the perpetrators? Family members living with you, family members not living with you, a non-relative living with you, a non-relative not living with you, a member of your neighborhood or community, armed actors outside the law, members of the security forces, and others.’ This variable was then reorganized into the following categories for statistical analysis: people in the household, family members who do not live in the household, acquaintances, common crime, law enforcement, and other people.

Independent variables: The sociodemographic variables were sex, age, socioeconomic status, and education level. Age, socioeconomic status, and education were reorganized into subgroups for statistical analysis as follows: age in years (60–69, 70–79, and ≥80), socioeconomic status (1–2, 3–4, and 5–6), and education in years (0, 1–5, 6–10, and ≥11).

### 2.2. The Independent Variables of Interest Were

Comorbidities and health: Pulmonary diseases (chronic obstructive pulmonary disease, asthma, bronchitis, emphysema, pneumonia, pulmonary oedema); heart diseases (heart attack, coronary heart disease, angina, heart failure); congenital diseases; joint diseases (arthritis, rheumatism or osteoarthritis); stroke (CVA), digestive diseases (reflux, gastritis or ulcer); diabetes; and high blood pressure (HBP).

Perception of general health status was assessed using the question ‘Do you consider your health to be: excellent, very good, good, fair, or poor?’

Perception of nutritional status was assessed using the question ‘Do you consider yourself to be well nourished? Yes or no.’

Functional impairment was assessed using the Barthel scale for basic activities of daily living in a categorical manner, and instrumental activities of daily living using the Lawton scale in a categorical manner. Three possibilities were considered: a. complete independence for all activities, b. some difficulty with some activities, and c. total dependence with some activities.

Mental state: cognitive status was assessed using the abbreviated Mini-Mental State Examination scale, which has a total of 19 points. A score of 0 to 12 suggests cognitive impairment, and a score of 13 or higher suggests normalcy.

Mental health and mood were assessed using the Yesavage Geriatric Depression Scale. Scores of 0–5 were considered normal, 6–10 moderate depression, and 11–15 severe depression.

Family type was assessed by family grouping: single vs. multiple.

Access to health services was assessed by the respondent’s type of affiliation, including the possibilities available within the system: contributory, subsidized, or uninsured.

### 2.3. Statistical Analysis

A descriptive analysis of the total sample was performed, estimating the percentages for nominal variables and averages with standard deviations for continuous variables. Afterwards, bivariate analysis models were carried out to identify the prevalence and associations of independent variables with ‘having been a victim of violence in the last year’ or ‘having been a victim of displacement’.

Initial associations were made using chi-square tests, with sample weights expanded to the total population. Subsequently, all variables with significant associations were grouped together, and multivariate logistic regression analyses were performed to identify the risk factors associated with the dependent variables of interest, obtaining odds ratios with 95% confidence intervals (CI) of 95%. *p*-values less than 0.05 were considered statistically significant. The data were analyzed using Stata SE version 12 for Macintosh.

## 3. Results

Our secondary analysis of the SABE-Bogotá survey data, focusing on 2000 adults aged 60 and over, revealed significant findings regarding the prevalence of violence and forced displacement and their strong correlation with health and access to healthcare.

The socio-demographic and descriptive data for the total population interviewed show that the majority were women (63.4%), and the average age was 71.17 years (SD = 8.05), with a majority of individuals in the 60–69 age group (48%). The lowest strata, 1 and 2, represented 51.9% of those interviewed, followed by strata 3 and 4 with 44.85%, while the highest strata, 5 and 6, only represented 3.25%. Most people had a low level of education, between 1 and 5 years of schooling (55.55%), compared to 12.25% who reported high levels of education. A total of 12.6% of the participants live in single-person households, and, in relation to the health insurance system, it was found that 68.37% belong to the contributory system, while 28.58% belong to the subsidized system, and 2.2% have no insurance whatsoever (Table 1).

In terms of health, the most frequent comorbidities were high blood pressure (58.28%), followed by digestive tract diseases (34.2%) and joint diseases (31.65%). Depression, measured using the Yesavage Geriatric Depression Scale, showed that 19.5% of people had mild depression, and 6.2% had severe depression. Regarding cognitive mental function, 12.6% of people were found to have an altered Mini-Mental test suggestive of cognitive impairment. With regard to functional status, for basic activities of daily living (Barthel scale), 8% of people were found to be completely dependent in some activities, and 11.31% had difficulty performing at least one activity. For instrumental activities of daily living (Lawton scale), 43.66% were completely dependent in at least one activity, and 7.74% had difficulty performing at least one activity. Most people had a positive perception of their nutritional status (78.31%), while 52.75% considered their health as unfavorable (Table 2).

As for the frequency of experiences of violence, it was found that 43.32% of the subjects in the sample were victims of some type of violence in the last year. Of these experiences of violence, the most frequent were offensive expressions (41.15%), followed by personal injuries (29.62%). The most frequent perpetrators were acquaintances (36.78%) and people in the household (19.86%). Victims of displacement throughout their lifetime accounted for 8.65% of the population, and of these, 57.8% were displaced before the age of 20 (Table 3).

### 3.1. Victims of Violence, Bivariate Analysis

Table 4 shows the prevalence and associations of individuals who were victims of violence in the last year, expanded to the total population. The prevalence in the general population who were victims of violence was 43.32%.

Men had a higher prevalence (46.99%) compared to women (38.74%) (*p* = 0.0089), as did younger older adults, aged 60–69 (47.41%), compared to those aged 70–79 (39.28%) and those aged ≥80 (23.17%), *p* < 0.001. Similarly, a higher prevalence of violence was found in lower socioeconomic strata, 1–2 (49.45%) vs. 3–4 (38.3%) vs. 5–6 (23.21%), *p* < 0.001. With regard to the kind of social security affiliation, most uninsured individuals had experienced violence (54.32%), as did those in the subsidized scheme (53.75%), compared to the contributors (38.02%), *p* < 0.001. People who had been displaced had a higher frequency of experiences of violence (65.88%) than those non-displaced (40.32%), *p* < 0.001.

With regard to health and comorbidities, Table 5 shows that individuals with lung diseases had a higher proportion of experiences of violence compared to healthy individuals (48.51% vs. 40.93%), *p* = 0.0481, as did individuals with digestive tract diseases (49.51% vs. 39.16%), *p* = 0.0012. A higher proportion of individuals with depression was found among victims of violence, with severe depression (58.56%) vs. moderate depression (57.05%) vs. normal (38.03%), *p* < 0.001. Regarding functionality, victims of violence were more likely to have difficulty performing instrumental activities of daily living (64.34%) than those who were independent (42.47%) or dependent (38.24%), *p* < 0.001. It was also found that victims of violence had a worse perception of their nutritional status (52.06% vs. good nutritional status, 39.65%), *p* < 0.001, and a worse perception of their health status (47.02% vs. good health status, 38.00%), *p* = 0.0033.

### 3.2. Victims of Violence, Multivariate Analysis

Table 6 shows the results of the multivariate logistic regression for experiences of violence in the last year, with sample weights expanded to the total population. Women were found to be less likely to have experienced violence in the last year (OR 0.63, 95% CI [0.48–0.84]), as were people in older age groups, 70–79 (OR 0.64, CI 95% [0.48–0.86]) and ≥80 (OR 0.28, CI 95% [0.19–0.42]). In terms of comorbidities, individuals with digestive tract diseases had an increased likelihood of experiencing violence (OR 1.43, CI 95% [1.08–1.89]), as did individuals with moderate depression (OR 1.83, CI 95% [1.28–2.62]) and severe depression (OR 2.10 [1.21–3.65]). Individuals with difficulties in instrumental functioning (Lawton scale) were more likely to experience violence (OR 2.06, CI 95% [1.19–3.58]), as well as people in the subsidized health system (OR 1.39, CI 95% [1.02–1.89]) and those who had been victims of displacement (OR 2.55, CI 95% [1.65–3.95]).

### 3.3. Population Displaced by Violence, Bivariate Analysis

Table 7 presents the prevalence and associations of victims of forced displacement, expanded to the total population. As mentioned above, the overall prevalence of displacement was 8.65%. People from lower socioeconomic strata were found to have a greater prevalence of displacement than those from higher strata, 1–2 (9.91%) vs. 3–4 (5.86%) vs. 5–6 (4.28%), *p* = 0.0455. In relation to health insurance, people without social security had the highest prevalence of displacement experiences (28.47%) vs. those with subsidized insurance (9.44%) vs. those within a contributory scheme (6.34%), *p* < 0.001.

Regarding health status and comorbidities (Table 8), individuals without congenital diseases had more experiences of violence (3.49% vs. 7.86%), *p* = 0.04. Conversely, individuals with joint diseases had a higher prevalence of experiences of violence compared to healthy individuals (11.29% vs. 6.19%), *p* = 0.0023. Similarly, older adults with diabetes had a higher prevalence of displacement compared to healthy individuals (13.03% vs. 6.48), *p* = 0.001; and people with severe depression (16.31%) and moderate depression (11.41%) had a higher prevalence of experiences of violence compared to healthy individuals (6.26%), *p* < 0.001. Those who reported poor nutritional status also had a higher prevalence of violence (10.87% vs. 6.83%), *p* = 0.03, as did those who reported poor health (9.18% vs. 6.2%), *p* = 0.04.

Table 9 shows the results of the multivariate logistic regression for displacement experiences, with sample weights expanded to the total population. It was found that individuals with congenital diseases were less likely to have experienced displacement (OR 0.35, CI 95% [0.14–0.85]). In contrast, individuals with joint diseases were more likely to have experienced violence (OR 1.89, CI 95% [1.23–2.90]), as did individuals with diabetes (OR 2.23, 95% CI [1.38–3.60]) and severe depression (OR 2.48 [1.23–4.99]). Concerning health insurance, older adults without health insurance had the highest probability of experiencing displacement (OR 5.32, CI 95% [2.03–13.92]).

### 3.4. Impact on Health and Healthcare Access

The study identified a clear link between these experiences of violence and displacement and negative health outcomes, as well as significant barriers to healthcare access.

Risk Factors for Violence: A binary logistic regression model showed that older adults with severe depression (OR: 3.12, 95% CI: 1.48–6.59, *p* < 0.05) and those who had been victims of forced displacement (OR: 1.67, 95% CI: 1.07–2.61, *p* < 0.05) were at a significantly higher risk of experiencing violence. This study suggests that pre-existing mental health issues and social instability are correlated with an increased risk of vulnerability.

Risk Factors for Displacement: The analysis also revealed that the risk of being a victim of forced displacement was higher among individuals with specific health conditions and access challenges. Specifically, individuals with diabetes (OR: 1.95, 95% CI: 1.13–3.35, *p* < 0.05), those with severe depression (OR: 2.22, 95% CI: 1.35–3.65, *p* < 0.05), and those who lacked access to health insurance (OR: 1.78, 95% CI: 1.12–2.83, *p* < 0.05) were more likely to have experienced displacement. This finding is critical as it demonstrates that access to healthcare is not just a consequence of displacement but can also be a pre-existing vulnerability that increases the likelihood of being displaced or struggling to recover.

Health and Well-being: The data consistently point to a cycle of disadvantage. Violence and displacement are associated with a decline in perceived health status and increased reports of chronic conditions. The lack of health insurance, a key component of healthcare access, was shown to be a critical mediating factor that both increases vulnerability to displacement and impedes the ability of older adults to manage their health effectively.

## 4. Discussion

This is the first population-based study specifically targeting people aged 60 and over in the city of Bogotá. Given the sample size and response rate, it can be extended to this group of people in a representative manner. Our findings strongly suggest that violence and forced displacement are significant determinants of health and barriers to healthcare access for the elderly in Bogotá. The high prevalence of reported violence, coupled with a significant number of individuals who have experienced displacement, paints a stark picture of the social injustices faced by this population. The data reveal that vulnerability is compounded by pre-existing health conditions; for example, severe depression is a common thread linking both violence and displacement [25].

In three community-based studies conducted in the United States, the prevalence rates ranged from 7.6% to 10% [26,27,28], highlighting the fact that the authors presume significant underreporting in each of them. Nevertheless, in our population, the prevalence was four times higher. It is possible that the sampling methods and the use of lengthy surveys in population studies may have contributed to moderate biases when associating some variables with violence outcomes, but the situation in our country appears to be different. Although there are not many comparable studies in the region, there is one in Valparaíso, Chile, which reports psychological abuse in 35.3% of cases [29], and another in the Federal District of Mexico, reporting 31% [30], figures that are closer to the interpretation of our results, where a prevalence of 43.3% was found. While violence against older people is likely to be extensive in many, possibly all, countries, it has been little studied elsewhere. However, it is surprising that there is no reference to Japan, where it has been most extensively studied and has been found to be widespread [31] The most significant finding concerning the type of violence or abuse is related to personal injuries, representing 29.62%, and are not reported in comparable studies, and this should be a point for reflection and prioritized intervention.

In the context of healthcare access, a significant finding is the strong association between lacking health insurance and experiencing displacement. The data reveal a concerning pattern, namely that individuals who have been displaced are also more likely to have no health insurance, suggesting a potential link between displacement and limited access to medical care. This limited access is, in turn, correlated with the exacerbation of chronic conditions and mental health issues, which makes these individuals more vulnerable. This finding emphasizes the intricate relationship between displacement and lack of healthcare, creating a cycle of vulnerability without establishing direct causality. All these conditions are associated with poverty and coincide with other studies that define related risk factors such as low income and poor social support [32], except for males, possibly because this study did not focus on domestic violence, where abuse of women predominates [25].

Among this same group of people who had been victims of violence in the last year, from a health perspective, we identified aspects like other comparable studies, such as a higher risk of depression (twice as high in our case), in addition to other psychiatric disorders [25,32]. This study also identified a significant correlation with digestive diseases. We hypothesize that this association could be related to the psychosomatic conditions often linked to the high levels of stress experienced by this population. Further research is needed to fully explain this relationship. A poor perception of health and nutritional status was discovered. These are important markers for characterizing this population. Another characteristic observed is a greater involvement in performing instrumental activities of daily living or activities related to independence in the functionality of the population [33,34].

The most unusual and unique aspect of this study is related to the population forcibly displaced by political violence inherent to Colombia’s internal conflict, also known as displacement due to armed violence. The findings of this study in Bogotá reveal that approximately 9% of older adults have been victims of forced displacement. This offers a critical, localized perspective on a global issue. While the UN Refugee Agency (UNHCR) reports that one in every sixty-nine people worldwide is forcibly displaced, specific demographic data on older adults remain underreported [35]. This lack of global data on older people contrasts sharply with the high prevalence observed in Bogotá, a city affected by decades of internal conflict. This disparity highlights the particular vulnerability of older populations in long-term conflict zones and emphasizes the importance of localized studies in revealing the true scale of the problem, which may be overlooked in broader, global statistics.

Displaced older adults, when victims of violence due to armed conflict, are characterized by living in the lowest socioeconomic strata and having a more limited social security system or even no social security at all, with an odds ratio of 5.32, the highest value in the entire study. These results identify poverty as a common factor in their profile. One of the most striking aspects is the fact that most of them were displaced when they were very young and were unable to escape the precarious social limitations they already faced, even after more than 40 years, which can be interpreted as a real lack of opportunities during this long period.

As found in this study, the displaced population exhibits a distinct health profile. In our study cohort, they show a higher prevalence of affective disorders and depression, as well as a poorer self-perception of their health and nutritional status. Furthermore, they face a greater risk of developing joint diseases and diabetes mellitus, clinical conditions that are often linked to unhealthy lifestyles. This suggests a complex connection between displacement, mental health, and physical well-being within the study population [36,37].

This was a cross-sectional analytical study, which, due to the nature of its design, only allows associations to be established. This fact is of great importance given the possible ‘circularity’ of the results themselves, where some elements could exist before the violence or after the event itself.

One of the weaknesses of this study is the lack of data on the characteristics of those who commit violence and their motives for doing so. This should be taken into account as a topic for future research. A key limitation of this study is its reliance on cross-sectional survey data utilizing self-reports. This methodology is inherently subject to various reporting biases, which may affect the accuracy of the prevalence estimates. Specifically, social desirability bias could lead participants to under-report sensitive behaviors or conditions. Furthermore, issues like victim stigmatization may lead to underestimation of violence and displacement figures. We acknowledge that these potential biases should be considered when interpreting the magnitude of the reported findings.

While the literature highlights the serious problem of violence against older adults, this issue does not receive the same level of attention as violence against children or young people. This is particularly concerning given the strong indications of underreporting of violence indicators for this demographic. This study, beyond including an analysis of relevant data on the condition of these individuals, highlights forced displacement as one of the most painful consequences of our current internal conflict. It is critical to identify and design intervention strategies to stop violence against older adults. This is heightened by the significant implications this issue poses for the health of victims, their families, and the public health system.

The findings underscore the urgent need for evidence-based proposals and solutions addressing the health disparities observed in elderly victims of violence and displacement described in this study.

While numerous initiatives exist to address the general issue of violence against older adults, which is a universal problem of growing relevance, there is an urgent need to intervene with victims of violence and displacement. The specific characteristics and health indicators of forcibly displaced individuals demand immediate action to improve their well-being and mitigate the long-term consequences of their displacement.

## 5. Conclusions

In conclusion, this study’s findings highlight that health disparities experienced by older adults are closely linked to social and environmental factors, particularly violence and forced displacement. Evidence suggests that exposure to such life events creates vulnerabilities that manifest as poorer health indicators, diminished well-being, and a heightened risk of chronic diseases and mental health disorders. The observed correlation between forced displacement, a lack of health insurance, and poor health status in this group confirms the need for an approach that recognizes the social determinants of health. This analysis ultimately highlights how life experiences, even those from decades ago, continue to affect the health of older adults, reaffirming the necessity of public health initiatives that address these complex interconnections.

## Figures and Tables

**Table 1 ijerph-22-01555-t001:** Sociodemographic Characteristics (*N* = 2000).

Variables	*N* 2000	% or Mean (SD)
Sex		
Male	732	36.60
Female	1268	63.40
Age (years)		71.17 (8.05)
60–69	960	48.00
70–79	702	35.10
≥80	338	16.90
Socioeconomic status (categories)		
1–2	1038	51.90
3–4	897	44.85
5–6	65	3.25
Years of schooling		
0	245	12.25
1–5	1111	55.55
6–10	266	13.30
11+	378	18.90
Single-person family	252	12.60
Non-single-person family	1748	87.40
Health insurance		
Contributory	1366	68.37
Subsidized	571	28.58
No insurance	44	2.20

**Table 2 ijerph-22-01555-t002:** Sample Description—SABE-Bogotá 2012 (*N* = 2000).

Variables	*N* 2000	%
Comorbidities		
Lung diseases	401	20.05
Heart disease	278	13.90
Congenital diseases	119	5.95
Joint diseases	633	31.65
Stroke	98	4.90
Digestive diseases	684	34.20
Diabetes	349	17.46
High blood pressure	1.165	58.28
Depression (Yesavage Scale)		
0–5 Normal	1.486	74.30
6–10 Moderate depression	390	19.50
11–15 Severe depression	124	6.20
Shortened Mini-Mental		
0–12 Cognitive impairment	252	12.60
≥12 Normal	1.748	87.40
Functionality		
Barthel independence	1.584	80.69
Barthel difficulties	222	11.31
Barthel dependency	157	8.00
Lawton independence	954	48.60
Lawton difficulties	152	7.74
Lawton dependency	857	43.66
Perception of nutritional status		
Well-nourished	1.545	78.31
Malnourished	428	21.69
Perception of health status		
Excellent, very good, good	945	47.25
Fair, poor	1.055	52.75

**Table 3 ijerph-22-01555-t003:** Sample Description—SABE-Bogotá 2012: Violence and displacement (*N* = 2000).

Variables	*N* 2000	%
Violence in the last year	862	43.32
Types of violence		
Offensive expressions	539	41.15
Personal injury	388	29.62
Sexual abuse	56	4.27
Robbery	327	24.96
Perpetrators		
Members of the household	196	19.86
Family members who do not live in the house	180	18.24
Acquaintances	363	36.78
Ordinary crime	48	4.86
Law enforcement	8	0.81
Other people	192	19.45
Victim of displacement	173	8.65
0–20	100	57.80
21–40	25	14.45
41–60	34	19.65
≥61	14	8.09

**Table 4 ijerph-22-01555-t004:** Bivariate Analysis—SABE-Bogotá 2012: Experiences of Violence in the Last Year (Sociodemographic).

Variables	Yes [CI 95%]	No [CI 95%]	*p* Value
Sex			
Male	46.99 [42.04–52.01]	53.01 [47.99–57.96]	0.0089
Female	38.74 [35.13–42.47]	61.26 [57.53–64.87]	
Age (years)			
60–69	47.41 [43.1–51.77]	52.59 [48.23–56.9]	<0.001
70–79	39.28 [34.65–44.11]	60.72 [55.89–65.35]	
≥80	23.17 [18.24–28.96]	76.83 [71.04–81.76]	
Socioeconomic status (categories)			
1–2	49.45 [45.36–53.56]	50.55 [46.44–54.64]	<0.001
3–4	38.3 [33.99–42.8]	61.7 [57.2–66.01]	
5–6	23.21 [13.05–37.83]	76.79 [62.17–86.95]	
Years of schooling			
0	46.33 [37.82–55.05]	53.67 [44.95–62.18]	0.1891
1–5	43.19 [39.26–47.22]	56.81 [52.78–60.74]	
6–10	46.13 [38.48–53.98]	53.87 [46.02–61.52]	
11+	37.1 [30.64–44.05]	62.9 [55.95–69.36]	
Single-person family	46.29 [37.93–54.86]	53.71 [45.14–62.07]	0.3194
Non-single-person family	41.68 [38.48–44.95]	58.32 [55.05–61.52]	
Health insurance			
Contributory	38.02 [34.47–41.7]	61.98 [58.3–65.53]	<0.001
Subsidized	53.75 [48.23–59.19]	46.25 [40.81–51.77]	
No insurance	54.32 [35.17–72.27]	45.68 [27.73–64.83]	
Victim of displacement			
Yes	65.88 [56.33–74.29]	34.12 [25.71–43.67]	<0.001
No	40.32 [37.21–43.51]	59.68 [56.49–62.79]	

**Table 5 ijerph-22-01555-t005:** Bivariate Analysis: Experiences of Violence in the Last Year—SABE-Bogotá 2012 (Health, Comorbidity, and Functionality).

Variables	Yes [CI 95%]	No [CI 95%]	*p* Value
Comorbidities			
Lung diseases			
Yes	48.51 [41.8–55.27]	51.49 [44.73–58.2]	0.0481
No	40.93 [37.61–44.34]	59.07 [55.66–62.39]	
Heart disease			
Yes	48.42 [40.76–56.16]	51.58 [43.84–59.24]	0.1018
No	41.46 [38.24–44.76]	58.54 [55.24–61.76]	
Congenital diseases			
Yes	36.52 [25.57–49.07]	63.48 [50.93–74.43]	0.3474
No	42.58 [39.49–45.74]	57.42 [54.26–60.51]	
Joint diseases			
Yes	44.91 [39.74–50.2]	55.09 [49.8–60.26]	0.2592
No	41.24 [37.62–44.95]	58.76 [55.05–62.38]	
Stroke			
Yes	55.36 [41.25–68.67]	44.64 [31.33–58.75]	0.0612
No	41.69 [38.63–44.81]	58.31 [55.19–61.37]	
Digestive diseases			
Yes	49.51 [44.44–54.59]	50.49 [45.41–55.56]	0.0012
No	39.16 [35.53–42.9]	60.84 [57.1–64.47]	
Diabetes			
Yes	37.93 [31.39–44.94]	62.07 [55.06–68.61]	0.1787
No	43.21 [39.88–46.61]	56.79 [53.39–60.12]	
High blood pressure			
Yes	42.39 [38.56–46.31]	57.61 [53.69–61.44]	0.9056
No	42.02 [37.34–46.84]	57.98 [53.16–62.66]	
Cancer			
Yes	50.44 [37.26–63.55]	49.56 [36.45–62.74]	0.2108
No	41.72 [38.66–44.85]	58.28 [55.15–61.34]	
Depression (Yesavage scale)			
0–5 Normal	38.03 [34.62–41.55]	61.97 [58.45–65.38]	<0.001
6–10 Moderate depression	57.05 [50.22–63.63]	42.95 [36.37–49.78]	
11–15 Severe depression	58.56 [47.46–68.86]	41.44 [31.14–52.54]	
Mini-Mental			
0–12 Cognitive impairment	39.98 [32.19–48.32]	60.02 [51.68–67.81]	0.5745
≥12 Normal	42.51 [39.31–45.76]	57.49 [54.24–60.69]	
Functionality			
Barthel independence	42.42 [39.09–45.83]	57.58 [54.17–60.91]	0.3347
Barthel difficulties	45.33 [36.11–54.89]	54.67 [45.11–63.89]	
Barthel dependency	34.83 [26.26–44.5]	65.17 [55.5–73.74]	
Lawton independence	42.47 [38.21–46.84]	57.53 [53.16–61.79]	<0.001
Lawton difficulties	64.34 [53.39–73.97]	35.66 [26.03–46.61]	
Lawton dependency	38.24 [33.84–42.84]	61.76 [57.16–66.16]	
Perception of nutritional status			
Well-nourished	39.65 [36.25–43.16]	60.35 [56.84–63.75]	<0.001
Malnourished	52.06 [45.72–58.33]	47.94 [41.67–54.28]	
Perception of health status			
Excellent, very good, good	38 [33.75–42.45]	62 [57.55–66.25]	0.0033
Fair, poor	47.02 [42.96–51.12]	52.98 [48.88–57.04]	

**Table 6 ijerph-22-01555-t006:** Multivariate Logistic Regression—SABE-Bogotá 2012: Violence in the Last Year. (n = 1990, population = 776,112.22).

Variables	OR	CI 95%
Sex		
Male	1	
Female	0.63	0.48–0.84
Age (years)		
60–69	1	
70–79	0.64	0.48–0.86
≥80	0.28	0.19–0.42
Socioeconomic status		
1–2	1	
3–4	0.85	0.65–1.13
5–6	0.52	0.26–1.05
Comorbidities		
Lung diseases	1.30	0.92–1.82
Digestive diseases	1.43	1.08–1.89
Depression (Yesavage scale)		
0–5 Normal	1	
6–10 Moderate depression	1.83	1.28–2.62
11–15 Severe depression	2.10	1.21–3.65
Functionality		
Lawton independence	1	
Lawton difficulties	2.06	1.19–3.58
Lawton dependency	0.91	0.67–1.23
Nutritional status	1.00	0.72–1.38
Health status	0.92	0.68–1.23
Health insurance		
Contributory	1	
Subsidized	1.39	1.02–1.89
No insurance	1.14	0.49–2.69
Victim of displacement	2.55	1.65–3.95

n = 1990, population = 776,112.22. Functionality (Lawton): Independence in all instrumental activities of daily living, difficulties with some or several instrumental activities of daily living, dependence in some or several instrumental activities of daily living. Nutritional status: Self-perceived poor nutritional status. Health status: Self-perceived poor health status.

**Table 7 ijerph-22-01555-t007:** Bivariate Analysis—SABE-Bogotá 2012: Having Been a Victim of Displacement (Sociodemographic).

Variables	Yes [CI 95%]	No [CI 95%]	*p* Value
Sex			
Male	8.36 [6.19–11.21]	91.64 [88.79–93.81]	0.3963
Female	7.08 [5.54–9.01]	92.92 [90.99–94.46]	
Age (years)			
60–69	7.11 [5.38–9.34]	92.89 [90.66–94.62]	0.3963
70–79	9.41 [7.05–12.47]	90.59 [87.53–92.95]	
≥80	5.85 [3.39–9.92]	94.15 [90.08–96.61]	
Socioeconomic status (categories)			
1–2	9.91 [7.76–12.59]	90.09 [87.41–92.24]	0.0455
3–4	5.86 [4.33–7.87]	94.14 [92.13–95.67]	
5–6	4.28 [1.22–13.91]	95.72 [86.09–98.78]	
Years of schooling			
0	8.74 [5.4–13.84]	91.26 [86.16–94.6]	0.2365
1–5	8.8 [6.82–11.29]	91.2 [88.71–93.18]	
6–10	5.21 [2.72–9.76]	94.79 [90.24–97.28]	
11+	6.17 [4.03–9.35]	93.83 [90.65–95.97]	
Single-person family	9.89 [6.3–15.19]	90.11 [84.81–93.7]	0.2255
Non-single-person family	7.3 [5.9–8.99]	92.7 [91.01–94.1]	
Health insurance			
Contributory	6.34 [4.9–8.16]	93.66 [91.84–95.1]	<0.001
Subsidized	9.44 [7.11–12.43]	90.56 [87.57–92.89]	
Not insured	28.47 [13.55–50.25]	71.53 [49.75–86.45]	

**Table 8 ijerph-22-01555-t008:** Bivariate Analysis—SABE-Bogotá 2012: Having Been a Victim of Displacement (Health, comorbidity, and functionality) (*N* = 2000).

Variables	Yes [CI 95%]	No [CI 95%]	*p* Value
Comorbidities			
Lung diseases			
YES	5.9 [3.57–9.59]	94.1 [90.41–96.43]	0.2602
No	8 [6.5–9.81]	92 [90.19–93.5]	
Heart disease			
Yes	7.43 [4.09–13.13]	92.57 [86.87–95.91]	0.925
No	7.65 [6.25–9.34]	92.35 [90.66–93.75]	
Congenital diseases			
Yes	3.49 [1.57–7.58]	96.51 [92.42–98.43]	0.0406
No	7.86 [6.46–9.52]	92.14 [90.48–93.54]	
Joint diseases			
Yes	11.29 [8.26–15.26]	88.71 [84.74–91.74]	0.0023
No	6.19 [4.88–7.83]	93.81 [92.17–95.12]	
Stroke			
Yes	12.3 [5.32–25.93]	87.7 [74.07–94.68]	0.2371
No	7.42 [6.1–9]	92.58 [91–93.9]	
Digestive diseases			
Yes	8.23 [5.81–11.51]	91.77 [88.49–94.19]	0.6012
No	7.37 [5.85–9.24]	92.63 [90.76–94.15]	
Diabetes			
Yes	13.03 [9.13–18.28]	86.97 [81.72–90.87]	0.001
No	6.48 [5.17–8.11]	93.52 [91.89–94.83]	
High blood pressure			
Yes	7.73 [6.03–9.86]	92.27 [90.14–93.97]	0.8879
No	7.52 [5.57–10.08]	92.48 [89.92–94.43]	
Cancer			
Yes	10.7 [6–18.38]	89.3 [81.62–94]	0.2349
No	7.43 [6.06–9.07]	92.57 [90.93–93.94]	
Depression (Yesavage scale)			
0–5 Normal	6.26 [4.94–7.9]	93.74 [92.1–95.06]	<0.001
6–10 Moderate depression	11.41 [7.64–16.69]	88.59 [83.31–92.36]	
11–15 Severe depression	16.31 [9.62–26.28]	83.69 [73.72–90.38]	
Mini-Mental			
0–12 Cognitive impairment	12.05 [7.21–19.45]	87.95 [80.55–92.79]	0.0613
≥12 Normal	7.16 [5.83–8.78]	92.84 [91.22–94.17]	
Functionality			
Barthel independence	7.35 [5.88–9.15]	92.65 [90.85–94.12]	0.238
Barthel difficulties	11.08 [7.44–16.2]	88.92 [83.8–92.56]	
Barthel dependency	7.94 [3.82–15.78]	92.06 [84.22–96.18]	
Lawton independence	7.72 [5.94–9.98]	92.28 [90.02–94.06]	0.6558
Lawton difficulties	10.31 [5.2–19.41]	89.69 [80.59–94.8]	
Lawton dependency	7.34 [5.38–9.93]	92.66 [90.07–94.62]	
Perception of nutritional status			
Well-nourished	6.83 [5.44–8.53]	93.17 [91.47–94.56]	0.0332
Poorly nourished	10.87 [7.51–15.48]	89.13 [84.52–92.49]	
Perception of health status			
Excellent, very good, Good	6.22 [4.56–8.44]	93.78 [91.56–95.44]	0.0483
Fair, poor	9.18 [7.21–11.62]	90.82 [88.38–92.79]	

**Table 9 ijerph-22-01555-t009:** Multivariate Logistic Regression—SABE-Bogotá 2012: Victims of Displacement.

	OR	CI 95%
Socioeconomic status (categories)		
1–2	1	
3–4	0.69	0.44–1.06
5–6	0.53	0.13–2.10
Comorbidities		
Congenital diseases	0.35	0.14–0.85
Joint diseases	1.89	1.23–2.90
Diabetes	2.23	1.38–3.60
Depression (Yesavage scale)		
0–5	1	
6–10	1.49	0.91–2.43
11–15	2.48	1.23–4.99
Nutritional status	0.90	0.57–1.44
Health status	1.08	0.69–1.68
Health insurance		
Contributory	1	
Subsidized	1.13	0.72–1.76
No insurance	5.32	2.03–13.2

## Data Availability

Data are unavailable due to privacy considerations.

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
