# Peer review of "Violence, Inequity, and Their Impact on Health and Access to Healthcare Services Among the Elderly Population of Bogotá"

_ijerph, 2025, doi:10.3390/ijerph22101555_

Round 1
Reviewer 1 Report
Comments and Suggestions for Authors
The authors correctly point out that violence against older people is likely to be extensive in many, possibly all, countries, but it has been little studied anywhere. However it is surprising that they do not refer to Japan where it has been most extensively studied and has been found to be widespread. References in English can be found in Mayumi Hayashi The Care of Older People (London: Pickering and Chatto, 2013). This also contains references to studies of elder abuse in Britain where it is not widely studied. The study of the victims of elder abuse in Bogota is original, methodologically sound, revealing and convincing. It provides very considerable detail about victims of abuse. As the authors suggest, it has the potential to inform improved arrangements for the protection and care of victims and potential victims. However it is a pity that they did not seek a comparable degree of detail about the perpetrators of violence and the circumstances in which it was inflicted upon the different socio-economic and age groups of older people. It is surely as important to seek to prevent and punish elder abuse as it is to care for older victims. More information about the circumstances of abuse could assist prevention and punishment in Colombia and elsewhere. Surprisingly this is not even mentioned as a desirable objective.
This article has much to commend it but it, and readers, would benefit from revision to take account of the above comments.
Author Response
Reviewer 1:
Comments 1:
The authors correctly point out that violence against older people is likely to be extensive in many, possibly all, countries, but it has been little studied anywhere. However it is surprising that they do not refer to Japan where it has been most extensively studied and has been found to be widespread. References in English can be found in Mayumi Hayashi The Care of Older People (London: Pickering and Chatto, 2013). This also contains references to studies of elder abuse in Britain where it is not widely studied.
Response 1: We agree with this comment. Therefore we include new reference and improve the discusión.
While violence against older people is likely to be extensive in many, possibly all, countries, it has been little studied elsewhere. However, it is surprising that there is no reference to Japan, where it has been most extensively studied and has been found to be widespread
Comments 2:
The study of the victims of elder abuse in Bogota is original, methodologically sound, revealing and convincing. It provides very considerable detail about victims of abuse. As the authors suggest, it has the potential to inform improved arrangements for the protection and care of victims and potential victims. However it is a pity that they did not seek a comparable degree of detail about the perpetrators of violence and the circumstances in which it was inflicted upon the different socio-economic and age groups of older people. It is surely as important to seek to prevent and punish elder abuse as it is to care for older victims. More information about the circumstances of abuse could assist prevention and punishment in Colombia and elsewhere. Surprisingly this is not even mentioned as a desirable objective.
Response 2:
Thank you for your valuable feedback. We acknowledge the importance of investigating the perpetrators of violence and the specific circumstances of their actions. However, our study's primary objective was to characterize the health profile and living conditions of older adult victims of violence and forced displacement. Given the specific context of Colombia's protracted internal conflict, our focus was on documenting the experiences and health consequences for this vulnerable population. Therefore, a detailed analysis of the perpetrators and circumstances of abuse, while a desirable objective for future research, fell outside the scope of the present work.
Reviewer 2 Report
Comments and Suggestions for Authors
Thank you for the opportunity to review your paper, which presents results of a secondary analysis of SABE-Bogota survey, exploring the association between self-reported experiences of violence, health and socio-economic factors in older population in Bogota. The paper is well written overall and provides some important insights. I have made a few suggestions in the introduction, discussion and conclusion. They focus on improving clarity, better dividing thoughts (currently some sentences are very long and present multiple thoughts), and strengthening discussion section with more dialogue with relevant literature, and evidence-based suggestions for strategies (currently strategies are only mentioned in conclusions).

Author Response
Response to reviewer 2
Comments 1. “It is difficult to understand what you mean with this sentence; please clarify and divide into 2 sentences”
The larger number of older adults not only contributed to a better understanding of their specific characteristics, but also to an awareness that they are attributed, as a group, 63 a series of conditions ranging from specific health situations to some very particular ones from a political, social and cultural perspective.
Response 1: We agree with this comment. Therefore, we divide into 2 sentences.
The increasing population of older adults has led to a more comprehensive understanding of their unique characteristics and needs. This demographic shift has also brought to light that this group is often subject to a range of attributions, from specific health conditions to unique political, social, and cultural circumstances. Understanding these multifaceted aspects is crucial for developing effective policies and interventions that cater to this growing population.
While recognizing the specificities of older adults is essential, there is a risk of generalizing and attributing a monolithic set of conditions to the entire group. These attributions, which can range from health-related vulnerabilities to political or social roles, can inadvertently perpetuate stereotypes. A critical academic perspective requires moving beyond these broad generalizations to acknowledge the diversity and heterogeneity within the older adult population. It's vital to recognize that an individual's experience is shaped by a complex interplay of personal history, socioeconomic status, and cultural background, rather than a predetermined set of group attributes.
Comments 2: “very long sentence, please divide”
In general, they document 85 recent and current events related to domestic violence in the household or in residential 86 care facilities, seeking to identify risk factors (13-16), and the information is collected from 87 adult social protection, public health or government agency records, such as the police, 88 emergency services and nursing homes in countries seeking to develop policies to deal 89 with this issue.
Response 2: We agree with this comment. Therefore, we rewritten the sentence.
Most often, they document recent and current events related to domestic violence in households or residential care facilities. They seek to identify risk factors (13–16) and collect information from adult social care, public health, or government agency records. These records may come from the police, emergency services, or nursing homes. Countries seeking to develop policies to deal with this issue can use this information..
Comments 3: unclear
“and given the current historical moment in our country,”
Response 3: We agree with this comment. Therefore, the wording was improved and the paragraph was rewritten.
Both clinicians and researchers face considerable challenges in identifying and addressing this issue. In this particular moment in our nation's history, it is not only necessary, but also of paramount importance, to thoroughly contextualise this problem.
Comments 4: please revise, it uses informal language and unclear what you mean
“to figure out how to get rid of poverty by using as 109 much money as possible, like from philanthropists and donors, and make sure that this 110 money has an impact on health and human development indicators”
Response 4: We agree with this comment. Therefore, the wording was improved and the paragraph was rewritten.
Since poverty is one of the triggers of violence, it is important to take into account the United Nations' proposals regarding the acquisition of resources to combat this phenomenon.
Comments 5: too long sentence that presents different ideas. please divide.
“This study explores the prevalence of violence and forced displacement as indicators of inequity among Bogotá's elderly population, with a particular focus on how these factors affect their health and access to healthcare services conducted by the Institute on Ageing of the Pontificia Universidad Javeriana, co-financed by the Administrative Department of Science, Technology and Innovation (Colciencias).”
Response 5: We agree with this comment. Therefore, we divide the ideas.
This study investigates the prevalence of violence and forced displacement as indicators of inequity within Bogotá's elderly population. It specifically examines how these factors impact their health and the way to access to healthcare services.
The research was conducted by the Institute on Ageing at the Pontificia Universidad Javeriana and was co-financed by the Administrative Department of Science, Technology and Innovation (Colciencias).
Comments 6: “unclear, how are you able to establish causality. your data comes from a cross-sectional survey, so we can only discuss association; not causality”
This highlights how pre-existing mental health issues and social instability act as powerful vulnerabilities.
Response 6: We agree with this comment. Therefore, the wording was improved.
This study suggests that pre-existing mental health issues and social instability are correlated with an increased risk of vulnerability.
Comments 7: “Your discussion is narrative in nature and you are engaging only little with available literature to show how your findings are similar/different from the literature. Also, please list some suggested/evidence-based types of strategies to address the key findings. Currently they are broadly mentioned only, in one sentence, in conclusions, which is not appropriate.”
Response 7: We agree with this comment. Therefore, the discussion was improved and rewritten.
Comments 8. While ther reader can appreciate the likely link between these social and structural determinants of health, it is unclear, how are you able to establish causality (as you describe it). your data comes from a cross-sectional survey, so we can only discuss association; not causality. please clarify.
Response 8: We agree with this comment. Therefore, we revised version of the paragraph that clarifies this distinction and presents the findings accurately.
In the context of healthcare access, a significant finding is the strong association between lacking health insurance and experiencing displacement. The data reveal a concerning pattern, namely that individuals who have been displaced are also more likely to have no health insurance, suggesting a potential link between displacement and limited access to medical care. This limited access is, in turn, correlated with the exacerbation of chronic conditions and mental health issues, which makes these individuals more vulnerable. This finding emphasizes the intricate relationship between displacement and a lack of healthcare, creating a cycle of vulnerability without establishing direct causality.
Comments 9: unclear sentence, could you please put it differently.
This study also identified a significant correlation with digestive diseases. We hypothesize that this association could be related to the psychosomatic conditions often linked to the high levels of stress experienced by this population. Further research is needed to fully explain this relationship.
Response 9: We agree with this comment. Therefore, we change the sentence.
This study also identified a significant correlation with digestive diseases. We hypothesize that this association could be related to the psychosomatic conditions often linked to the high levels of stress experienced by this population. Further research is needed to fully explain this relationship.
Comments 10: not grammatically correct, please revise
Similarly, a poor perception of health and nutritional status has been found, which are elements that contribute as markers for the characterization of this population.
Respond 10: We agree with this comment. Therefore, The paragraph was rewritten.
A poor perception of health and nutritional status has been discovered. These are important markers for characterizing this population.
Comments 11: do you mean, in your study? please clarify
Nearly 9 out of 100 older adult individuals in the city of Bogotá have been victims of this type of violence, and what is most striking is that most 426 of them were young when it happened, 6 out of 10, which is not surprising in a conflict that has now lasted more than 50 years.
Respond 11: We agree with this comment. Therefore, we clarify and attatched a reference.
The findings of this Bogotá-based study, which reveal that around 9% of older adults have experienced forced displacement, provide a vital localised perspective on a global issue. While the UN Refugee Agency (UNHCR) reports that one in every 69 people worldwide are forcibly displaced, specific demographic data on older adults remains underreported. The high prevalence observed in Bogotá, a city impacted by decades of internal conflict, contrasts sharply with this lack of global data on the elderly. This disparity highlights the unique vulnerability of older populations in protracted conflict zones and emphasises the importance of localised studies in revealing the true extent of the problem, which may be overlooked in broader, aggregated global statistics.
Comments 12: please clarify throughout the discussion which sentences describe your study population, by specifying "as found in this study...." otherwise unclear
Respond 12: We agree with this comment. Therefore, we clarify throughout the discussion using “as found in this study.
As found in this study, the displaced population exhibits a distinct health profile. In our study cohort, they show a higher prevalence of affective disorders and depression, as well as a poorer self-perception of their health and nutritional status. Furthermore, they face a greater risk of developing joint diseases and diabetes mellitus, clinical conditions that are often linked to unhealthy lifestyles. This suggests a complex connection between displacement, mental health, and physical well-being within the study population.
Comments 13: consider shortening: this study used a cross-sectional study, which....
Respond 13: We agree with this comment. Therefore, we change the sentence.
This study used a cross-sectional analytical study,
Comments 14: please clarify what you mean, currently the sentence uncler
Respond 14: We agree with this comment. Therefore, we rewritten the paragraf.
The findings of this study in Bogotá reveal that approximately 9% of older adults have been victims of forced displacement. This offers a critical, localised perspective on a global issue. While the UN Refugee Agency (UNHCR) reports that one in every 69 people worldwide are forcibly displaced, specific demographic data on older adults remains underreported [35]. This lack of global data on older people contrasts sharply with the high prevalence observed in Bogotá, a city affected by decades of internal conflict. This disparity highlights the particular vulnerability of older populations in long-term conflict zones and emphasizes the importance of localized studies in revealing the true scale of the problem, which may be overlooked in broader, global statistics.
Comment 15: please clarify what do you mean: it is important to identify strategies to addresss...
Respond 15: We agree with this comment. Therefore we rewritten the sentence.
It is mandatory to identify and design intervention strategies to stop violence against older adults. This is heightened by the significant implications this issue poses for the health of victims, their families and the public health system.
Comment 16: too long sentence, contains multiple ideas. Please divide, and revise to make it clearer. For example consider replacing "we are behind in proposing solutions and interviews" with "strategies are urgently needed, to address...."
Respond 16: We agree with this comment. Therefore we divided in two sentences.
We can no longer put off the proposals and solutions required by the elderly victims of both violence and displacement described in this study.
While numerous initiatives exist to address the general issue of violence against older adults, which is a universal problem of growing relevance, there is an urgent need to intervene with the second group. The specific characteristics and health indicators of forcibly displaced individuals demand immediate action to improve their well-being and mitigate the long-term consequences of their displacement.
Comment 17: these proposed strategies are not discussed in discussion. they should be discussed, and referenced there. please don't include new information (that was not included in results and/or discussion) in conclusions
Respond 17: we agree with this commnet. Therefore, we rewritten the conclusion.
In conclusion, this study's findings highlight that health disparities experienced by older adults are closely linked to social and environmental factors, particularly violence and forced displacement. Evidence suggests that exposure to such life events creates vulnerabilities that manifest as poorer health indicators, diminished well-being and a heightened risk of chronic diseases and mental health disorders. The observed correlation between forced displacement, a lack of health insurance and poor health status in this group confirms the need for an approach that recognises the social determinants of health. This analysis ultimately highlights how life experiences, even those from decades ago, continue to affect the health of older adults, reaffirming the necessity of public health initiatives that address these complex interconnections.
Round 2
Reviewer 1 Report
Comments and Suggestions for Authors
This article is much improved and is particularly helpful in describing the characteristics of older people who are particularly vulnerable to violence. It is still a weakness that there is no discussion of the characteristics of those who commit violence and their motives. This would be helpful for those seeking to prevent violence. Perhaps no evidence is available. If so this should be pointed out and the need for more information about perpetrators to be the subject of future research.
Comments on the Quality of English Language
In general the English is good but the final sentence of paragraph 5 , page 2, is unclear and should be revised. It states: '...having in mind to avoid perpetuating stereotypes'. 'the need' should be inserted between 'mind' and 'to avoid'.
Author Response
Comment 1:
“This article is much improved and is particularly helpful in describing the characteristics of older people who are particularly vulnerable to violence. It is still a weakness that there is no discussion of the characteristics of those who commit violence and their motives. This would be helpful for those seeking to prevent violence. Perhaps no evidence is available. If so this should be pointed out and the need for more information about perpetrators to be the subject of future research.”
Respond 1:
We agree with this comment. Therefore we include new sentence:
One of the weaknesses of this study is the lack of data on the characteristics of those who commit violence and their motives for doing so. This should be taken into account as a topic for future research.
Comments on the Quality of English Language
Comment 2:
In general the English is good but the final sentence of paragraph 5 , page 2, is unclear and should be revised. It states: '...having in mind to avoid perpetuating stereotypes'. 'the need' should be inserted between 'mind' and 'to avoid'.
We agree with this comment. Therefore we change the sentence:
"It is crucial to understand these diverse aspects in order to develop effective policies and interventions having in mind the need to avoid perpetuating stereotypes."